Analysing head and trunk motion in the judo osoto-gari technique: relationship to sweeping-leg velocity

Liu Lingjun 1 llj190220@sina.com
Deguchi Tatsuya 2
Shiokawa Mitsuhisa 3
http://orcid.org/0009-0008-1129-1284 Hamaguchi Kazuto 2
Shinya Masahiro 2
1 Shanghai Research Institute of Sports Science (Shanghai Anti-Doping Agency) , Shanghai , China
2 Graduate School of Humanities and Social Sciences, Hiroshima University , Higashi-Hiroshima , Japan
3 Faculty of Health and Sports Sciences, Hiroshima International University , Higashi-Hiroshima , Japan
Federolf Peter
Electronic publication date: 2025 Jan 23
Publication date: 2025
Volume: 13
Electronic Location ID: e18862
Received 2024 Oct 4; Accepted 2024 Dec 23
Copyright: © 2025 Liu et al.
Copyright year: 2025
Copyright holder: Liu et al.
License: This is an open access article distributed under the terms of the Creative Commons Attribution License, which permits unrestricted use, distribution, reproduction and adaptation in any medium and for any purpose provided that it is properly attributed. For attribution, the original author(s), title, publication source (PeerJ) and either DOI or URL of the article must be cited.
License URL: https://creativecommons.org/licenses/by/4.0/

Keywords: Judo biomechanics, Motion analysis, Kinematics, Sports performance, Correlation analysis

Funding: The authors received no funding for this work.

==============================
Background

Osoto-gari is a leg throw technique that primarily relies on the hip extension to initiate the sweeping motion of the leg. A high sweep contact velocity is a crucial factor in efficiently executing this technique. While some literature emphasises whole-body coordination in the leg-sweeping action, the roles of trunk and head motion remain unclear. This study investigates head and trunk movements (including the pelvic and upper torso) contributing to higher leg-sweep velocities when executing the judo osoto-gari.

Methods

Kinematic data were collected from 17 male black-belt judokas using a motion capture system (250 Hz). Pearson product-moment correlation and stepwise linear regression were used to identify kinematic variables linked to the sweeping-leg velocity at sweep contact (SC).

Results

Six out of twenty-four variables correlated with sweeping-leg velocity at SC. A stepwise regression model (adjusted R2 = 0.53, p = 0.009) predicted sweeping-leg velocity based on head-tilt angle at maximum sweeping-leg height (MSH) and SC, head-tilt angular velocity at MSH, and trunk-tilt angular velocity at MSH.

Conclusions

The findings of this study indicate that (1) increasing the forward angle of the head aids the visual system in rapidly processing spatial information about the target position, thus facilitating the execution of the leg sweep, and (2) a greater forward-tilt rotation of the head, which leads to rapid trunk rotation, is conducive to enhancing sweeping-leg velocity.

Introduction

In judo, throwing techniques are often named based on the body parts crucial to their execution (Imamura & Johnson, 2003; Matsumoto, 1975). Among these, osoto-gari stands out as a leg throw primarily leveraging hip extension to initiate a sweeping motion. This sweeping action targets the outside of the opponent’s leg, propelling their upper body backward towards the ground (Fig. 1). Executing a powerful leg sweep is essential for osoto-gari, which can momentarily lift the opponent’s feet off the ground, allowing the thrower to easily drive their upper body backward towards the ground (Liu et al., 2022). Therefore, studying the mechanisms of the leg-sweep action, encompassing biomechanical principles, is critical for an in-depth understanding of how to optimise and improve the leg-sweep technique, as well as for enhancing the throwing efficiency of osoto-gari.

Figure 1 Sequential photography of osoto-gari.

The thrower is on the left and the faller is on the right. The thrower takes a large step forward with the pivot (left) leg and raises the sweeping (right) leg, as shown in 1–4. The thrower then makes sweeping contact with the faller’s outside (right) leg and pushes his upper body backward towards the mat (5–8). Images 4(a) and 8(a) are front views of the photographs shown in 4 and 8, respectively.

Previous biomechanical studies have examined the leg-sweeping motion in osoto-gari, shedding light on key aspects of its execution. For example, Imamura & Johnson (2003) highlighted the significance of the sweeping leg in its execution, aligning with the kinetic chain principle. They found that, compared to white-belt judokas (novices), black-belt judokas (skilled judokas) demonstrated higher peak velocities, particularly with the plantar flexion position, as they swept their leg to contact the opponent’s leg. Similarly, Liu et al. (2021) investigated biomechanical differences in osoto-gari moves for various levels of judokas. They observed that black-belt judokas tended to initiate the upward-swinging motion of the thigh and shank later than white-belt judokas. Furthermore, during the backward-sweep phase, the black-belt judokas moved their thighs at a larger angular velocity. Melo et al. (2012) explored kinematic variations in the sweeping-leg motion when fallers (that is, opponents being thrown) of different heights were thrown. The results revealed that the knee and hip extension angles of the sweeping-leg motion were greater when the thrower threw a shorter faller. While these insights have enhanced our understanding of the fundamental mechanics required to effectively execute the osoto-gari leg-sweep, the significance of whole-body coordination during the execution of the movement should not be overlooked.

Most judo throwing techniques involve whole-body movement, requiring precise coordination of muscular strength and joint motion involving the arms, legs, and trunk (Matsumoto, 1975; Scott, 2019). For example, Choi & Song (2023) demonstrated that compared with non-elite judokas, elite judokas exhibit significantly larger shoulder, pelvic, hip, and knee angles at the moment of maximal pulling force during seoi-nage. These findings highlight the superior coordination of upper and lower body movements in elite athletes, enabling better whole-body control. Additionally, Ishii et al. (2019) reported higher peak velocities in the ankle and hip joints during the turn phase among elite judokas, with these velocities being positively correlated with the peak angular momentum. This suggests that rapid, coordinated lower-limb movement drives angular momentum, contributing to a more powerful and efficient seoi-nage throw. Building on this understanding of whole-body dynamics, a recent investigation into osoto-gari, Liu et al. (2022) examined the correlation between the supporting-leg kinetics and leg-sweep velocity during the throw. They found that the external moment of the centre of mass generated by the external force of the pivot leg correlated with the leg-sweep velocity during the sweep contact. This finding suggests that an increase in the external moment induces rapid forward rotation of the trunk in the throwing direction; this, in turn, drives the accelerated rotation of the sweeping leg, thereby accelerating the leg sweep. While the pivot leg kinetics play a crucial role in effectively executing the osoto-gari leg sweep, it is important not to disregard the significance of upper body motion during the execution of the movement.

Some judo coaching literature (Daigo, 2005; Yamashita, 1992) emphasises the importance of upper body-to-upper body impact during the osoto-gari throw. This emphasis aligns with studies that highlighted the role of trunk and upper-body coordination in generating the momentum necessary for successful techniques. Research into osoto-gari throwing has revealed that the thrower’s rapid tilting of the trunk forward can generate a larger momentum, resulting in a stronger push from the upper body (Imamura & Johnson, 2003). Similarly, Liu et al. (2020) reported that from a biomechanical perspective, during osoto-gari execution, upper-body-to-upper-body contact reduces the external moment arm (the distance between the body’s centre of gravity and the centre of pressure) and increases the external moment. This adjustment can increase the thrower’s angular momentum, contributing to improved throwing effectiveness. Nevertheless, it remains to be seen whether upper body movements can influence the velocity of the leg sweep.

The head may play a significant role in the execution of judo-throwing techniques. Yamashita (1992) emphasizes that during the leg sweep in osoto-gari, the head should be pressed downward to ensure effective execution. Despite this guidance in coaching literature, there is a lack of biomechanical studies investigating the specific contributions of head motion in osoto-gari. From a motor control perspective, head motion is involved in visual feedback and influences body balance. For example, Magnani et al. (2020) demonstrated that variations in head orientation significantly affect gait stability during walking, highlighting the interplay between head position and postural stability. Because the osoto-gari leg sweep action is performed with one leg supporting the body, thus placing the athlete in a relatively unstable posture, we posit that tilting the head forward during the throwing phase (Fig. 1) may influence the stability of the body. This adjustment could facilitate balance control, enhance coordination between the trunk and limbs, and ultimately improve the effectiveness of the throw.

Therefore, to gain deeper insight into the potential impact of head and trunk motions (including the pelvic and upper torso) on leg-sweep velocity in osoto-gari, we aimed to examine the relationship between these motions and leg-sweep velocity exhibited by black-belt judokas. We tested two hypotheses: (1) The angles and angular velocities of head tilt, trunk tilt, upper torso, and pelvic motions at specific points in time are significantly correlated with leg-sweep velocity at the moment of sweep contact (SC; i.e., when the thrower’s sweeping leg makes contact with the faller’s outer leg). (2) There exist several crucial correlations that can accurately predict leg-sweep velocity at SC. By investigating these relationships, we seek to provide a more comprehensive understanding of the mechanisms underlying the osoto-gari leg sweep, offering valuable insights for coaches and judokas seeking to enhance osoto-gari performance.

Materials and Methods

Participants

Seventeen male university judokas (age: 20.5 ± 2.4 years; height: 172.8 ± 4.3 cm; mass: 82.6 ± 13.4 kg; judo experience: 12.1 ± 3.3 years) served as the thrower. Additionally, one male 2nd-degree black-belt judoka (age: 22 years; height: 173 cm; mass: 81 kg) served as the faller. The faller’s height and mass were selected to be representative of average values to ensure that all throwers could effectively execute the throws. All judokas voluntarily participated in this study. All the throwers were black belts in Judo, holding at least a 1st-degree rank according to the Japanese Judo grading system. They possessed regional and national-level competition experience, proficiently executing osoto-gari throws in practices or competitions. Notably, the participants, both throwers and faller, had not experienced any serious musculoskeletal injuries in the 3 months preceding data collection. We conducted a post hoc power analysis for the correlation tests using G* Power (version 3.1.9.7). For the variables that exhibited significant correlations (refer to Tables 1 and 2), we confirmed that the majority achieved a statistical power (1-β) exceeding 0.80. This study was approved by the ethics committee of Hiroshima University (approval number: 01–33). The testing protocol was explained to each participant, and all participants provided written informed consent.

Table 1 Angle variables and their correlations with the sweeping-leg velocity at SC.

Kinematic variables (°)	Mean ± SD	r	p	95% CI	
Head-tilt angle at SLL	20.55 ± 10.46	0.000	0.999	[−0.480 to 0.481]	
Head-tilt angle at MSH	40.57 ± 13.38	−0.609	0.009**	[−0.843 to −0.182]	
Head-tilt angle at SC	58.78 ± 16.79	−0.490	0.046*	[−0.785 to −0.012]	
Trunk-tilt angle at SLL	−6.07 ± 6.85	−0.231	0.372	[−0.641 to 0.281]	
Trunk-tilt angle at MSH	7.81 ± 12.23	−0.592	0.012*	[−0.835 to −0.156]	
Trunk-tilt angle at SC	29.10 ± 15.94	−0.335	0.189	[−0.703 to 0.174]	
Upper torso angle at SLL	−28.35 ± 12.02	0.284	0.269	[−0.227 to 0.673]	
Upper torso angle at MSH	−37.70 ± 14.97	0.349	0.169	[−0.158 to 0.711]	
Upper torso angle at SC	−15.93 ± 26.68	0.276	0.283	[−0.236 to 0.668]	
Pelvic angle at SLL	−5.85 ± 8.38	−0.113	0.666	[−0.563 to 0.389]	
Pelvic angle at MSH	14.27 ± 14.18	−0.073	0.780	[−0.535 to 0.422]	
Pelvic angle at SC	30.02 ± 15.49	0.034	0.898	[−0.454 to 0.506]	
Notes:

Mean ± SD.

* p < 0.05.

** p < 0.01.

SLL: the instant of the sweeping leg leaving the ground; MSH: the instant of maximum sweeping leg height; SC: the instant of sweep contact. The bold entries indicates the variable is significantly correlated with sweeping-leg velocity.

Table 2 Angular velocity variables and their correlations with the sweeping-leg velocity at SC.

Kinematic variables (°/s)	Mean ± SD	r	p	95% CI	
Head-tilt angular velocity at SLL	63.87 ± 64.45	−0.696	0.002**	[−0.882 to −0.324]	
Head-tilt angular velocity at MSH	121.41 ± 73.54	−0.619	0.008**	[−0.847 to −0.196]	
Head-tilt angular velocity at SC	79.43 ± 57.82	−0.011	0.968	[−0.489 to 0.472]	
Trunk-tilt angular velocity at SLL	11.66 ± 35.03	−0.176	0.499	[−0.605 to 0.333]	
Trunk-tilt angular velocity at MSH	109.83 ± 62.54	−0.670	0.003**	[−0.870 to −0.280]	
Trunk-tilt angular velocity at SC	151.11 ± 43.24	−0.045	0.863	[−0.515 to 0.445]	
Upper torso angular velocity at SLL	−74.53 ± 41.38	0.274	0.287	[−0.238 to 0.667]	
Upper torso angular velocity at MSH	34.01 ± 54.80	−0.120	0.645	[−0.568 to 0.382]	
Upper torso angular velocity at SC	249.28 ± 141.42	−0.301	0.241	[−0.683 to 0.210]	
Pelvic angular velocity at SLL	81.55 ± 66.38	−0.026	0.920	[−0.501 to 0.460]	
Pelvic angular velocity at MSH	111.99 ± 66.04	0.430	0.085	[−0.063 to 0.755]	
Pelvic angular velocity at SC	73.01 ± 76.67	−0.469	0.057	[−0.775 to 0.015]	
Notes:

Mean ± SD.

** p < 0.01.

SLL: the instant of the sweeping leg leaving the ground; MSH: the instant of maximum sweeping leg height; SC: the instant of sweep contact. The bold entries indicates the variable is significantly correlated with sweeping-leg velocity.

Measurements

Figure 2 illustrates the experimental setup, wherein 47 reflective markers were attached to the bodies of both thrower and faller. These markers were positioned at specific anatomical areas: head vertex, left/right ear, acromion, anterior/posterior shoulders, suprasternal notch, xiphoid process, seventh cervical, 10th thoracic vertebrae, lateral coastal borders, third metacarpal of the hands, medial/lateral wrist joints, medial/lateral elbow joints, left/right toe tip, first and fifth metatarsals, calcaneal tuberosities, medial/lateral ankle joints, medial/lateral knee joints, left/right greater trochanter, and anterior/superior iliac spines. Following a warm-up, the participants were allowed to practice osoto-gari to adjust themselves to the measurement environment. The faller wore specially designed judo gear (comprising only sleeves and a collar), and assumed a standing posture without resisting the throwers’ movements. The throwers were instructed to execute the osoto-gari move with maximum effort.

Figure 2 Measurement setup.

In studying the complexity of the osoto-gari technique, we adopted a dual selection method that integrates both subjective and objective assessments. This approach has been utilised in previous judo research (Ishii et al., 2018; Liu et al., 2021). After each throw, the throwers subjectively evaluated their athletic performance using a five-point Likert scale (1 = poor, 2 = below average, 3 = average, 4 = good, 5 = excellent). Each thrower repeated the throw twice or until a score of 4 or 5 was successfully captured. Three experienced university judo coaches (coaching experience: 23, 30, and 40 years) independently judged and selected the best performance for each participant, which was used for subsequent data analysis. Three-dimensional (3D) marker trajectories were recorded by a 14-camera Mac3D motion analysis system (Motion Analysis Corp., Santa Rosa, CA, USA) operating at a frequency of 250 Hz.

Data processing

All data processing was performed using MATLAB 2016a (MathWorks Inc., Natick, MA, USA). The 3D marker trajectory data were smoothed using a fourth-order low-pass Butterworth filter. The cut-off frequency was set to 12 Hz (Liu et al., 2021, 2022). The analysis phase of the osoto-gari motion was from the instant the thrower’s pivot leg was lifted off the ground to the point at which a body part of the faller contacted the judo mat (that is, the vertical GRF of the faller exceeded 5 N). The key instants, sweeping leg leaving the ground (SLL), maximum sweeping-leg height (MSH), and sweep contact (SC), were identified using kinematic and kinetic data collected throughout the osoto-gari motion. SLL was defined as the instant the vertical ground reaction force of the thrower’s sweeping leg was below 5 N. MSH was defined as the instant the thrower’s sweeping leg toe marker was at the highest vertical position. SC was determined following the method outlined in a previous study (Liu et al., 2022), defined as the instant at which the peak horizontal velocity (opposite to the throwing direction) of the sweeping leg’s ankle joint markers occurred. Osoto-gari was divided into two phases: swing and throw (Liu et al., 2021). The first phase spanned from the instant at which the thrower’s pivot leg left the ground to the MSH, while the second phase encompassed the period from MSH to the instant the thrower’s body contacted the ground. Each phase was normalised to 100%.

The head-tilt angle was determined as the angle formed by the line connecting the midpoint between the xiphoid process and the 10th thoracic vertebra, the top of the head marker, and the Z-axis in the YZ plane. The trunk-tilt angle was defined as the angle formed by the line connecting the midpoint between the right/left anterior iliac spines, the midpoint between the right/left shoulder joint centre, and the Z-axis in the YZ plane. The upper torso and pelvic angles were defined as the lines connecting the right/left shoulder joint centre, right/left anterior iliac spines, and the Y-axis in the XY plane, respectively. All calculated kinematic data for both the left- and right-handed participants were adjusted to represent each participant as right-handed.

Each motion direction was defined as follows. Head tilt and trunk tilt: forward (+) tilt (in the throwing direction)/backward (−) tilt (opposite to the throwing direction); upper torso and pelvic: forward (+) rotation (throwing direction)/backward (−) rotation (opposite to the throwing direction). The kinematic variables (angle and angular velocity) at SLL, MSH and SC, as illustrated by Figs. 3 and 4, were extracted for subsequent analysis of their correlation with the sweeping-leg velocity at SC.

Figure 3 (A–D) Mean (standard deviation) time-series data for the angles of the head, trunk, upper torso and pelvic of the thrower.

SLL: the instant of the sweeping leg leaving the ground; MSH: the instant of maximum sweeping leg height; SC: the instant of sweep contact.

Figure 4 (A–D) Mean (standard deviation) time-series data for the angular velocities of the head, trunk, upper torso and pelvic of the thrower.

SLL: the instant of the sweeping leg leaving the ground; MSH: the instant of maximum sweeping leg height; SC: the instant of sweep contact.

Statistical analysis

All statistical analyses were conducted using JASP 0.14.1.0, a freely available statistical software package (JASP Team, University of Amsterdam, Amsterdam, The Netherlands, 2020). All calculated data were presented as mean ± standard deviation (SD). The Pearson product-moment correlation coefficient (r) was employed to determine the relationships between the selected kinematic variables and peak sweeping-leg velocity. The alpha level for the statistical tests was set at p = 0.05. Correlation coefficient r values of 0.1, 0.3, 0.5, 0.7, and 0.9 were adopted as the thresholds for low, moderate, large, very large, and extremely large correlations, respectively (Hopkins et al., 2009). Subsequently, a stepwise multiple regression analysis was performed, in which sweeping-leg velocity was designated as the dependent variable. According to the methodology described by Fuchs et al. (2019), only those angle and angular velocity variables that demonstrated significant correlations were selected as independent variables. To assess multicollinearity in the regression model, the Pearson correlation coefficients (r) were first calculated to evaluate the relationships between the variables. Correlations were considered problematic if r ≥ 0.90 (Richardson, Mitchell & Hughes, 2016). Subsequently, the variance inflation factor (VIF) and tolerance values were computed. Multicollinearity was considered a concern if the VIF values exceeded 10 or the tolerance values were below 0.1 (O’Brien, 2007). The threshold for statistical significance in the regression analysis was set at p = 0.05.

Results

The sweeping-leg velocity at SC was –4.16 ± 0.97 m/s (the negative value corresponds to the direction opposite to the throwing direction).

Figure 3 presents the angles for the head-tilt, trunk-tilt, upper torso, and pelvic rotations. When the thrower swept their leg off the ground, both the head (Fig. 3A) and trunk (Fig. 3B) continuously tilted in the forward direction (+). The head-tilt angles at the MSH (40.57 ± 13.38°) (r = –0.609, p = 0.009, 95% CI [−0.843 to −0.182], larger negative) and SC (58.78 ± 16.79°) (r = −0.490, p = 0.046, 95% CI [−0.785 to −0.012], larger negative), and the trunk-tilt angle at MSH (7.81 ± 12.23°) (r = –0.592, p = 0.012, 95% CI [−0.835 to −0.156], larger negative) were significantly correlated with the sweeping-leg velocity at SC, as listed in Table 1. The angles of the upper torso (Fig. 3C) and pelvis (Fig. 3D) presented very similar patterns. At the beginning of the motion, the upper torso and pelvis continuously rotated in the backward direction (−), and they peaked during the phase between the MSH and SC. Subsequently, the upper torso and trunk twist rotated in the forward direction (+). However, no considerable correlations were found in the upper torso and pelvic rotation angles (Table 1).

Figure 4 displays the angular velocities for the head-tilt, trunk-tilt, upper torso, and pelvic rotations. From the 50% normalised time, the head (Fig. 4A) tilted rapidly in the forward direction (+), peaking near the MSH; thereafter, it continuously decreased, while the trunk (Fig. 4B) continually tilted in the forward direction (+). The head-tilt angular velocity at SLL (63.87 ± 64.45°/s) (r = −0.696, p = 0.002, 95% CI [−0.882 to −0.324], larger negative) and MSH (121.41 ± 73.54°/s) (r = –0.619, p = 0.008, 95% CI [−0.847 to −0.196], larger negative), and the trunk-tilt angular velocity at MSH (109.83 ± 62.54°/s) (r = –0.670, p = 0.003, 95% CI [−0.870 to −0.280], larger negative) were significantly correlated with the sweeping-leg velocity at SC, as shown in Table 2. However, no significant correlations were found in the upper torso (Fig. 4C) and pelvis (Fig. 4D) (Table 2).

Among the twenty-four independent variables examined, six were found to be correlated with sweeping-leg velocity at SC, as detailed in Tables 1 and 2. The regression model for predicting sweeping-leg velocity included the following variables: head-tilt angle at the MSH, head-tilt angle at SC, head-tilt angular velocity at MSH, and trunk-tilt angular velocity at MSH, as listed in Table 3. This model explained 53.5% of the variance in sweeping-leg velocity (R2 = 0.651, adjusted R2 = 0.535) and demonstrated a statistically significant fit (F (4, 12) = 5.605, p = 0.009). Among the predictor variables, the head-tilt angular velocity at MSH showed a significant negative association with sweeping-leg velocity (B = −0.012, t = −2.471, p = 0.029).

Table 3 Model of regression between predictor variables and the sweeping leg velocity at SC.

Predictor variables	β	t	p	
Constant	−3.749	–	–	
Head tilt angle at MSH	−0.043	−1.758	0.104	
Head tilt angle at SC	0.056	2.155	0.052	
Head tilt angular velocity at MSH	−0.012	−2.471	0.029*	
Trunk tilt angular velocity at MSH	−0.005	−1.048	0.315	
R 2	0.651	
Adjusted R²	0.535	
F	F (4, 12) = 5.605, **p = 0.009	
Notes:

Dependent variable: sweeping-leg velocity at SC. β: Unstandardized coefficients.

* p < 0.05.

** p < 0.01.

MSH: the instant of maximum sweeping leg height; SC: the instant of sweep contact.

Discussion

The primary aim of this study was to examine the relationship between head and trunk motions and sweeping-leg velocity in black-belt judokas. The results revealed significant correlations between head-tilt angles at MSH and SC, as well as the trunk-tilt angle at MSH and sweeping-leg velocity at SC, as shown in Table 1. Additionally, the angular velocities of head tilt at SLL and MSH, along with trunk tilt at MSH, were significantly correlated with sweeping-leg velocity at SC, as depicted in Table 2. Thus, the hypothesis of this study was partially supported. Regression analysis indicated that the independent variables (head-tilt angle at MSH and SC, head-tilt angular velocity at MSH, and trunk-tilt angular velocity at MSH) explained 53% of the variation in predicted sweeping-leg velocity (Table 3). This model suggests that head and trunk motions have a significant impact on leg-sweep velocity.

Following the thrower’s SLL, the head gradually tilted forward (+) (Fig. 3A), indicating that the black-belt judokas maintain a forward-leaning head posture throughout the throw. The statistical results show that the head-tilt angles at the MSH and SC were strongly correlated with the sweeping-leg velocity at SC (Table 1). In addition, the head-tilt angle at SC emerged as a predictive variable in the regression analysis, with a p-value of 0.052, which approached statistical significance (Table 3). Yamashita (1992) suggested that the thrower’s head should be pressed down against the opponent’s arm during the throw to force the faller’s upper body to tilt backward, preventing a counterattack. However, it is challenging to explain why the head-tilt angle correlates with the leg-sweep velocity, as there is limited literature on the influence of the head on osoto-gari. Motor control studies (Thomas et al., 2020; Tanaka et al., 2022) have indicated that the position of the head, as the primary receptor of visual and vestibular input, is crucial for maintaining body stability and executing motor tasks effectively. Based on these findings, we conjectured that tilting the head downwards may help maintain body balance while providing a clearer view, which enhances the accuracy of the leg-sweep motion, as illustrated in Fig. 1 (panel 4a).

Some judo experts (Daigo, 2005; Imamura & Johnson, 2003; Yamashita, 1992) emphasise the importance of precise leg contact towards the back of the opponent’s knee to maximise the effectiveness of a leg sweep. Similar to hitting sports such as baseball (Kishita, Ueda & Kashino, 2020; Higuchi et al., 2018) and table tennis (Shinkai et al., 2022), athletes must control their head position to align with their eye scan, enabling quick transmission of target information to the brain through the visual system for rapid action execution. Thus, tilting the head forward likely creates a clear line of sight for executing leg sweeping, indirectly enhancing the speed of the leg sweep.

Notably, both the angular velocities of head forward tilt (Fig. 4A) and trunk forward tilt (Fig. 4B) at MSH strongly correlated with the leg-sweep velocity at SC (Table 2). Certain judo experts (Daigo, 2005; Yamashita, 1992) have emphasised a strong upper contact for effectively executing osoto-gari. Additionally, some studies (Imamura & Johnson, 2003; Melo et al., 2012) indicate that a faster angular velocity of the thrower’s trunk generates greater angular momentum, facilitating a more efficient throw. Additionally, Liu et al. (2022) suggest that as the trunk rotates forward during a throw, it can drive the rotation of the sweeping leg, contributing to an increase in sweeping-leg velocity.

Among the predictors analysed, the head-tilt angular velocity at MSH was identified as a significant factor influencing the sweeping-leg velocity (p = 0.029) (Table 3). The regression analysis indicates that a 1 m/s increase in the sweeping-leg velocity corresponds to an approximately 80°/s increase in the head-tilt angular velocity at MSH. This suggests that an increase in the head-tilt angular velocity directly contributes to a higher sweeping-leg velocity. During MSH, the forward tilt of the head creates a “jackknife” effect (Fig. 1, panels 4 and 4(a)), which not only lifts the sweeping leg higher but also brings the head’s mass closer to the body’s axis of rotation. This forward tilt effectively reduces the body’s moment of inertia, enabling faster rotation. Consequently, as the thrower’s sweeping leg moves backwards (Fig. 1, panels 5–6), the entire body rotates more rapidly in the direction of the throw, thereby increasing the angular momentum. The sweeping action in osoto-gari is a multi-joint, coordinated movement involving the thigh, shank, and foot (Imamura & Johnson, 2003; Liu et al., 2022). Generating a greater angular momentum and transmitting it to the sweeping leg is essential for increasing the velocity of the sweeping leg and enhancing the effectiveness of the throw.

A biomechanical study of the golf downswing (Horan & Kavanagh, 2012) found a strong coupling between the rotation speed of the head and thorax, suggesting that controlling head rotation helps synchronize upper-body movements for an optimal swing. Similarly, in judo osoto-gari, head rotation is transmitted through the cervical spine to the thoracic and lumbar regions, initiating trunk rotation. Our results show that the head-tilt velocity begins to increase at 40% of the normalized time, whereas the trunk-tilt velocity starts to increase at 60%. By the time of MSH, the angular velocities of the head (121.41 ± 73.54°/s) and trunk (109.83 ± 62.54°/s) were nearly identical. This coupling further promotes the “jackknife” effect, which enhances the height of the sweeping leg and reduces the body’s moment of inertia during MSH, facilitating rapid rotation in the direction of the throw. Thus, black belts rapidly tilt their heads forward during the swing phase, driving rapid trunk rotation. This coordination seems beneficial for the leg-sweeping action as well, suggesting that head motion positively influences leg-sweep velocity.

The present study offers some practical implications for training and teaching. Coaches should emphasise maintaining an optimal head position, particularly when the sweeping leg is poised to move backward; tilting the head forward ensures a clear line of sight, enhancing the velocity and accuracy of the sweeping motion. Additionally, coordination between head and trunk movements is vital. Engaging the neck flexor and trunk flexor muscles simultaneously facilitates rapid trunk rotation, thereby increasing the velocity of the sweeping leg.

There are some limitations to this study. Firstly, the results are based on the coordinated physical abilities and technical proficiency of black-belt judokas, which may not apply to judokas of different levels. Secondly, our data analysis relied on a single successful trial per participant, and despite the involvement of three experienced judo coaches in the subjective evaluation process, there remains a possibility of selection bias. Nonetheless, this study highlights the importance of trunk and head movements on leg sweeping during the execution of the osoto-gari technique through rigorous biomechanical analysis. These insights can contribute to the enhancement of teaching and learning methods in osoto-gari.

Conclusions

The linear regression model developed in this study successfully predicted sweeping-leg velocity (R2 = 0.53) based on head-tilt angle at maximum MSH and SC, head-tilt angular velocity at MSH, and trunk-tilt angular velocity at MSH. This finding suggests that increasing the forward angle of the head facilitates the visual system’s rapid processing of spatial information regarding the target position, thereby enabling quicker execution of the leg sweep. Furthermore, a greater forward-tilt rotation of the head, leading to rapid trunk rotation, is beneficial for enhancing sweeping-leg velocity at the point of contact. These discoveries highlight the significance of controlling head and trunk rotations in effectively performing the leg sweeping motion.

This knowledge can significantly benefit coaches and judo athletes by providing valuable insights into applying optimal osoto-gari leg-sweeping techniques. Understanding the intricate relationship between head and trunk motions and leg-sweep velocity can enhance training methods and improve performance in judo competitions.

Supplemental Information

Supplemental Information 1 Raw data for Table 1.

Supplemental Information 2 Raw data for Table 2.

The authors thank the participants for participating in this study.

Additional Information and Declarations

Competing Interests

The authors declare that they have no competing interests.

Author Contributions

Lingjun Liu conceived and designed the experiments, performed the experiments, analyzed the data, prepared figures and/or tables, authored or reviewed drafts of the article, and approved the final draft.

Tatsuya Deguchi performed the experiments, authored or reviewed drafts of the article, and approved the final draft.

Mitsuhisa Shiokawa performed the experiments, authored or reviewed drafts of the article, and approved the final draft.

Kazuto Hamaguchi performed the experiments, authored or reviewed drafts of the article, and approved the final draft.

Masahiro Shinya conceived and designed the experiments, performed the experiments, authored or reviewed drafts of the article, and approved the final draft.

Human Ethics

The following information was supplied relating to ethical approvals (i.e., approving body and any reference numbers):

Hiroshima University granted Ethics review subcommittee of Integrated Arts and Sciences approval (approval number: 01–33).

Data Availability

The following information was supplied regarding data availability:

The raw angle and angular veloctiy data are available in the Supplemental Files.

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
