# Peer review of "Analysing head and trunk motion in the judo osoto-gari technique: relationship to sweeping-leg velocity"

_PeerJ, doi:10.7717/peerj.18862_

## Round 0.1 · original submission · Minor Revisions

The reviewers made several reasonable suggestions. Please consider them.

·

Basic reporting

Please include a few more recent studies (not older than 5 years) in the introduction part and in the discussion of the research problem.

Support advice from coaching literature (Matsumoto, 1975 - line 69; Yamashita 1992 and Daigo 2005 - line 79; Yamashita 1992 - line 87) with findings from relevant scientific research.

Experimental design

Please include information on the statistical strength of the sample in the "Participants" section (line 101-112).

Validity of the findings

No comment

Additional comments

No comment

Reviewer 2 ·

Basic reporting

General comments
This manuscript aims to investigate head and trunk movements (including the pelvic and upper torso) contributing to higher leg-sweep velocities when executing the judo osoto-gari. The aim is commendable. Authors found that increasing the forward angle of the head aids the visual system in rapidly processing spatial information about the target position, thus facilitating the execution of the leg sweep, and a greater forward-tilt rotation of the head, which leads to rapid trunk rotation, is conducive to enhancing sweeping-leg velocity. Overall, the authors manage to fulfil their aim sufficiently.

Experimental design

no comment

Validity of the findings

no comment

Additional comments

no comment

·

Basic reporting

The language is clear and the text is well structured.

Experimental design

The study is well designed, with good methodological quality and clear results. I have only minor suggestions:
- Line 102: Were the athletes in the same weight class? This is important in judo throwing techniques.
- Line 177-181: Did the authors test for multicollinearity in the data (multiple regression analysis)? Measures of VIF, correlation, and tolerance are fundamental, as it is normal for some variables to be correlated with each other.

Validity of the findings

- Line 213-216: I suggest that instead of showing the prediction equation (which is not useful), show the R2, adjusted R2, indicator (variables that entered into the model), standardized coefficients (β) and p in a table.
- Line 255-257: Our body is articulated, so the trunk and head play an important role in the osoto-gari technique, as their greater inclination allows the lower limb to reach a greater height and produce greater angular velocity and consequently greater angular momentum (lower moment of inertia), similar to the movement of a 'jackknife'. Please expand this discussion on angular momentum, moment of inertia and angular velocity. Perhaps, showing the equations to explain it.

---

## Round 0.2 · accepted · Accept

i m happy with the current version of the manuscript.